# Changing Exposure Perceptions: A Randomized Controlled Trial of an Intervention with Smoking Parents

**DOI:** 10.3390/ijerph17103349

**Published:** 2020-05-12

**Authors:** Vicki Myers, Shoshana Shiloh, David M. Zucker, Laura J. Rosen

**Affiliations:** 1Department of Health Promotion, School of Public Health, Sackler Faculty of Medicine, Tel Aviv University, Ramat Aviv, Tel Aviv 6997801, Israel; rosenl@tauex.tau.ac.il; 2School of Psychological Sciences, Gershon H. Gordon Faculty of Social Sciences, Tel Aviv University, Ramat Aviv, Tel Aviv 6997801, Israel; shoshi@tauex.tau.ac.il; 3Department of Statistics, Hebrew University, Mount Scopus, Jerusalem 9190501, Israel; david.zucker@mail.huji.ac.il

**Keywords:** tobacco smoke exposure, children, RCT, parental smoking, secondhand smoke, intervention, motivational interviewing, perceptions

## Abstract

Children who live with smokers are at risk of poor health, and of becoming smokers themselves. Misperceptions of the nature of tobacco smoke exposure have been demonstrated among parents, resulting in continued smoking in their children’s environment. This study aimed to change parents’ perceptions of exposure by providing information on second- and third-hand exposure and personalised information on children’s exposure [NIH registry (NCT02867241)]. One hundred and fifty-nine families with a child < 8 years and at least one smoking parent were randomized into intervention (69), control (70), and enhanced control (20) groups. Reported exposure, parental smoking details, and a child hair sample were obtained at the start of the study and 6–8 months later. Parental perceptions of exposure (PPE) were assessed via a questionnaire. The intervention consisted of motivational interviews, feedback of home air quality and child’s hair nicotine level, and information brochures. PPE were significantly higher at the study end (94.6 ± 17.6) compared to study beginning (86.5 ± 19.3) in intervention and enhanced control groups (t(72) = −3.950; *p* < 0.001). PPE at study end were significantly higher in the intervention group compared to the regular control group (*p* = 0.020). There was no significant interaction between time and group. Parallel changes in parental smoking behaviour were found. Parental perceptions of exposure were increased significantly post intervention, indicating that they can be altered. By making parents more aware of exposure and the circumstances in which it occurs, we can help parents change their smoking behaviour and better protect their children.

## 1. Introduction

Exposure to tobacco smoke causes a litany of health consequences with small children being particularly susceptible due to their small size and frequent contact with surfaces and caretakers. Consistent evidence has linked second-hand smoke in children with chronic ear infections, respiratory infections, sudden infant death syndrome, and impaired lung and heart function [1,2]. Since an estimated 40% of children worldwide are exposed to tobacco smoke on a regular basis [3], mostly due to living with smokers, the resulting health burden is significant. Children of smoking parents are further at higher risk of becoming smokers themselves [4].

Many parents smoke in or around the home, or in their child’s presence, without being aware of exposure occurring, often considering their protective behaviours to be sufficient to prevent exposure occurring [5]. Partial home bans are sometimes used by parents in an attempt to protect children, for example restricting smoking to certain areas, or refraining from smoking when children are present; however, this is often insufficient to prevent exposure from occurring [6,7,8].

Several interventions have attempted to reduce children’s exposure using various methods to provide information to parents, including motivational interviewing, self-help materials [9,10], face-to-face or telephone counselling [11,12], and biomarker feedback [11]; and in different settings, including the home [9,10] and healthcare centres [11,12,13]. Previous interventions also targeted different age groups, for example babies [12], children aged up to age 4 [9,14], up to age 12 [10,11], or up to 17 years old [15,16]. Reviews of such interventions have showed mixed effects, with little benefit to intervention participants in some cases, often due to concurrent changes in intervention and control groups [17,18]. 

Qualitative research with smoking parents demonstrated variance in awareness and perceptions of children’s exposure to tobacco smoke, and indicated that sensory and physical factors may influence parents’ perceptions of exposure [5]. Parental misconceptions about children’s exposure to smoke may therefore be a potential risk factor for parents smoking in the presence of children. A questionnaire was developed and validated to quantify Parental Perceptions of Exposure, rating exposure via pictures and vignettes: smokers rated lower exposure than non-smokers, indicating that they were less aware of exposure [19]. Higher perceptions of exposure were also found to be associated with less parental smoking around children [20].

We therefore designed an intervention with smoking parents that would specifically target parental perceptions of exposure (PPE). The current study aimed to change parental perceptions of exposure via a home-based intervention with smoking parents of young children, who spend the most time at home and in proximity to their parents. The purpose of the study was to assess the effect of the intervention on PPE and smoking behaviour among smoking parents enrolled in a randomised controlled trial (RCT). 

## 2. Materials and Methods 

Ethical approval was received from the Assaf Harofeh Hospital Helsinki committee and from Tel Aviv University Ethics’ committee. The trial was registered with the NIH clinical trials registry (NCT02867241). Participants provided informed consent both for their own and for their child’s participation in the study.

**Recruitment**: Families were recruited initially through Naamat daycare centres and subsequently via parent groups on social media (Facebook) in Israel, as well as via snowball sampling procedures. 

**Inclusion criteria** were families with one or two smoking parent(s), who smoke(s) at least 10 cigarettes per week, a child up to age 8, willingness to provide 2 hair samples and child’s hair length of at least 3 cm, availability for the next 6 months for follow-up, living in the study area. Each family received a gift voucher worth around $65 at the end of the study. Reasons for non-participation included having a child of the wrong age, having recently quit smoking, unwillingness to cut the child’s hair, smoking only nargila (water pipe) or smoking less than 10 cigarettes per week, and living in areas too far away for home visits.

**Design:** The study’s protocol is outlined in Figure 1. The intervention lasted six months.

Participants in the intervention group received a total of 4 home visits, 2 brochures with information on tobacco smoke exposure (developed and produced in-house) of which one was specifically designed to challenge misconceptions surrounding children’s exposure, 2 motivational interviews (during home visits), the option to conduct a home air test using a Sidepak or Dylos machine, feedback of their children’s hair nicotine result at the third meeting, and access to the study Facebook group where relevant material was posted. The motivational interviews culminated in setting a realistic goal for change to reduce children’s exposure. Parents completed the PPE questionnaire at the study beginning and end, and provided reported exposure data at each meeting. 

The enhanced control group was included to evaluate the “mere measurement effect”, the effect of asking questions about a behaviour on its performance [21], without further intervention. Participants in this group received a total of two home visits. In the first visit, they were asked to complete the PPE questionnaire. At the second and last visit, they received their children’s hair nicotine result, two brochures with information on tobacco smoke exposure, and concurrently completed the final PPE questionnaire and provided reported exposure data.

Participants in the regular control group received the same protocol as the enhanced control group, except that they did not complete the PPE at baseline. 

All final visits took place approximately 6–8 months after the first visit.

**Measurement:** Study data were obtained at home visits made by trained interviewers. 

PPE: Parental perceptions of exposure were assessed by questionnaire that parents completed online. The PPE questionnaire consists of 23 items: 8 photographs and 9 text vignettes, describing adults smoking around children in various circumstances, including in the home, car, and outdoors, or specified amounts of time after smoking in the home and car. The full questionnaire can be found in Appendix A. Participants are asked: “to what extent is the described child exposed to tobacco smoke (to what extent does the smoke reach him/her)?” on a scale of 1 to 7 for each item. Items are summed to give a total score, and divided by the number of items to give a mean score. Higher scores denote a broader definition of exposure and rating children as more exposed in hypothetical situations where tobacco smoke is present. Six further questions relate to perceived knowledge and were summed to give a perceived knowledge score. See [19] for more details on contents, development, and validation of the PPE. PPE change scores (T2–T1) were computed among participants who filled out the questionnaire twice, at the beginning and end of the study.

Reported exposure was assessed by interview with parents, the interviewer completing a questionnaire. This included parental smoking status, number of cigarettes smoked per day, home smoking location, frequency of smoking around the child at home and away from the home, intent to quit and general health questions. Parents also provided socio-demographic data. At the final visit, parents were asked if they believed the child’s exposure level had changed over the last six months, and if so, to give the reason (open question). These reasons were then classified into the following categories of behavioural change: quit smoking (one or both parents)/reduced smoking/implemented a home or car smoking ban/smoke less around children/close balcony door when smoking/greater awareness of exposure.

Biofeedback of children’s exposure was assessed via hair nicotine—a hair sample was obtained from one child in each family at study beginning and end (both samples from the same child). All families provided a hair sample at the study beginning. Hair analysis was conducted by the Johns Hopkins University secondhand smoke exposure lab, which used gas chromatography with mass spectrometer detection (GC/MS), and provided nicotine values in ng/mg and whether the sample was above or below the limit of detection (LOD). In the current study, hair nicotine was used purely for feedback to parents.

**Statistics**: Normality was assessed and PPE scores were close to normal distribution. Paired t tests were used to determine whether there was a significant change in PPE between the study beginning and end in the Intervention and Enhanced control groups. Independent t tests were used to determine differences between PPE at the study end between intervention and regular control groups; and two-way ANOVA was used to compare intervention and enhanced control groups at the study end controlling for baseline. The relationship between perceived knowledge at baseline and change in PPE at study end was assessed by Pearson correlation. Analyses were conducted in SPSS version 25. 

**Programme adherence:** 100% of intervention families received the first motivational interview (at home); 98% of intervention families received the second motivational interview; 70% of intervention families had home air quality testing. The study was completed by 97% of families (68/69 in the intervention group; 67/70 in the control group; 18/20 in the enhanced control group); PPE at study end was available for 91% of the families.

## 3. Results

Three hundred and ninety-two families expressed initial interest in the study, of which we were able to contact 357; of these, 80 were unsuitable based on inclusion criteria, and a further 117 declined to participate. One family was randomized in error and subsequently withdrew, leaving a total of 159 families who participated in the RCT, and were randomized into 3 groups. Participants’ characteristics are presented in Table 1. Sixty-two percent of the mothers and 87% of the fathers were smokers. The groups did not differ in their baseline demographic characteristics.

### 3.1. Parental Perceptions of Exposure (PPE)

Paired t tests showed that PPE scores were significantly higher at study end (T2) compared to the study beginning (T1) in both the intervention and enhanced control groups who completed the questionnaire twice (Table 2). Mean PPE change from study beginning to end was not significantly different between the two groups (independent t test *p* = 0.2). 

Independent t-test comparing the intervention and regular control groups at study end showed the intervention group had significantly higher PPE at study end (F = 2.234, *p* = 0.020) (Figure 2). There was not a significant difference at study end between intervention and enhanced control groups.

Two-way ANOVA examining the effect of group and time in the intervention and enhanced control groups showed a significant effect of time (F = 20.99, *p* < 0.001) but no significant effect of group (F = 0.098, *p* = 0.755) and no significant interaction between time and group (F = 0.201; *p* = 0.655).

Those who rated lower perceived knowledge at baseline showed greater increase in PPE at study end (r = −0.326, *p* = 0.008).

### 3.2. Behavioural Outcomes

Changes in parental smoking behaviour were reported by the study end, with several parents reporting having quit smoking completely, or stopped smoking in the home, or reported smoking less around their children and making more effort to take protective measures (Table 3). Significantly more intervention families (39%) reported some positive change compared to control families (17% regular control, 10% enhanced control) (*p* = 0.003). Families (in the intervention and enhanced control groups) who made any positive behavioural change showed a greater change in PPE score (n = 24; mean = 11.3; SD = 15.8) than those who did not make a change (n = 49; mean = 6.5; SD = 18.2), though this difference was not statistically significant (t = 1.09; *p* = 0.28). 

## 4. Discussion

A randomized controlled trial of an intervention with smoking parents showed that parental perceptions of exposure increased significantly from the start to end of the trial in both intervention and enhanced control groups, and were significantly higher at study end in the intervention group compared to the regular control group. 

These findings indicate that parents’ perceptions of exposure are changeable, an optimistic discovery considering the link between parents’ perceptions of exposure and their smoking behaviours in their children’s environment [11,12]. This concurs with the increasing recognition that interventions which aim to change inaccurate or unhelpful perceptions are important practices for health promotion [22]. The intervention used in our study to modify parents’ perceptions of exposure combined targeted information to challenge common misperceptions about exposure to tobacco smoke, and personal feedback of household air quality and of a biomarker of children’s exposure. This combination of methods was designed to make parents aware of exposure, where they may have been previously unaware. The use of pictures (provided in both the PPE questionnaire and the brochure) may be especially helpful in providing parents with information about the nature of exposure and the circumstances in which it occurs. 

Interestingly, the enhanced control group, who completed the PPE questionnaire at the beginning of the study without receiving the intervention, also showed significant increases in perceptions of exposure at follow-up. Moreover, comparing the intervention and the enhanced control groups showed that their changes in exposure perceptions did not differ significantly. Both groups completed the PPE questionnaire at the beginning of the study and subsequently showed increases in perceptions of exposure. The ‘mere measurement’ effect is commonly regarded as an artifact, threatening the validity of assessments of intervention effectiveness [23]. Unlike this study which targeted perceptions, most of the literature on the ‘mere measurement’ effect deals with question–behavior effects (QBE), how simply answering questions about a specific behaviour may change that behaviour. Findings indicate small effect sizes of QBE with considerable heterogeneity between studies [24]. However, other studies, focusing on cognitions (perceptions, beliefs, attitudes), have shown that using questionnaires can result in participants forming beliefs about topics to which they have previously devoted little thought, possibly by increasing the salience or accessibility of beliefs about specific aspects of performing a health behaviour [25,26,27,28]. Correspondingly, some participants in our study commented that completing the questionnaire made them think about where they smoke and about their children’s exposure. While tempting to suggest that filling out the PPE be considered as an intervention in itself (much simpler and cheaper than the extended intervention), one must be cautious. This study was not designed to test the mere effects of filling out the questionnaire, and the sample size of the enhanced control group was not powered accordingly. However, the findings call for further investigation of the effects of using the PPE measure per se as an intervention. 

Parallel to changes in parental perceptions of exposure, changes in parentally reported smoking behaviour were also seen, with significantly more intervention families (39% compared to 10–17% of control families) reporting any behavioural change to reduce children’s exposure to smoke. This compares with other interventions with smoking parents. For example, Abdullah [29] reported that 15% quit in the intervention group compared to 7% in the control group with telephone counselling; while others found no significant benefit to the intervention group, e.g. Eriksen [30] who provided counselling and information brochures to parents. These studies did not, however, look at perceptions. Other interventions that have aimed to change perceptions—such as illness perceptions in patients following myocardial infarction [31] and in those suffering lower back pain [32]—reported changes in illness perceptions and parallel positive changes in recovery outcomes. In the case of illness perceptions, actions taken to reduce health risks were proposed to be guided by the individual’s subjective or common-sense constructions of the health threat [33]. With regards to exposure perceptions, which have not been explicitly targeted by previous interventions, smoking behaviour around children is proposed to be influenced by parents’ subjective constructions regarding exposure. 

Finally, those who rated lower perceived knowledge at baseline also showed greater increase in PPE at the study end, perhaps indicating that awareness of lack of knowledge on the subject increases willingness to accept new information about exposure. These findings suggest that smoking parents with low perceived exposure (those who are less inclined to perceive exposure as occurring) and/or low perceived knowledge on the topic are the most appropriate target for an intervention focusing on children’s exposure to smoke. 

There were some limitations to the study design, including potential selection bias since participants volunteered, limiting the sample to those who were willing to take part in the research, potentially those who are healthier or more affluent, although participants were compensated for their time [34]. Participants may also have been more concerned about exposure or have a greater desire to test their child’s exposure compared to non-participants. The sample was largely secular and of average or above average SES, with low representation of religious and low SES populations, although participants were recruited from different geographic locations around the country in an attempt to increase diversity. Very young children were excluded if they did not have enough hair for a sample. Recruitment was limited geographically due to logistical restraints. 

It should be noted that behavioural outcomes were self-reported, and quitting smoking was not biochemically verified. These results could have been influenced by social desirability bias or a desire to please the interviewer. Furthermore, it is important to note that the enhanced control group had only 20 participants, of which only 12 had complete data at both time-points, compared to 61 participants with complete data in the intervention group, potentially limiting the power to detect a significant effect. Since PPE was not measured in the regular control group at baseline, we could not assess change in perceptions in this group.

We cannot isolate which element of the intervention had an effect on PPE—the motivational interviews including making a written plan of action, the brochures, the hair feedback, or simply trial participation. However, since there were increases in both intervention and enhanced control groups, we can assume that the elements common to both groups—trial participation including providing a hair sample and completion of the PPE questionnaire—were important in contributing to changing perceptions of exposure.

## 5. Conclusions

This study, using a specially designed intervention including interviews, written materials, and biomarker feedback, was the first to explicitly target parental perceptions of exposure. The intervention attempted to demonstrate children’s smoke exposure to parents in smoking households. By the end of the study, exposure perception scores had increased in both intervention and enhanced control groups. The potential of mere measurement effects on consequent changes in perceptions and behaviours, while promising, deserve further investigation. Parental smoking behaviour also changed by study end, with a higher percentage of intervention families reporting some positive change in parental smoking behaviour. By making parents more aware of exposure and the circumstances in which it occurs, we can help to protect children from the dangers of tobacco smoke.

## Figures and Tables

**Figure 1 ijerph-17-03349-f001:**
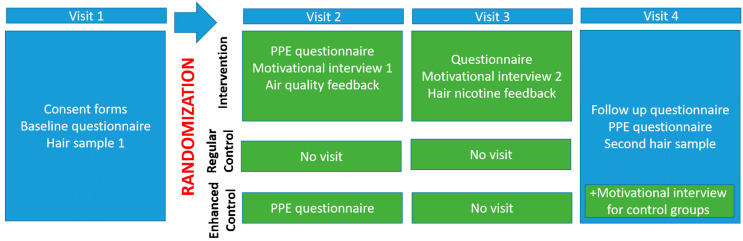
Study protocol by group.

**Figure 2 ijerph-17-03349-f002:**
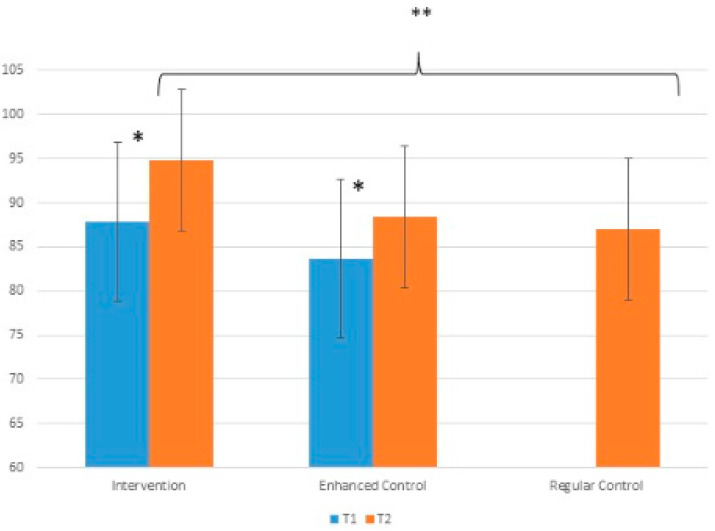
PPE over time across study groups. * denominates significant difference (*p* < 0.05) between T1 and T2. ** denominates significant difference (*p* < 0.05) between intervention and regular control at T2. Error bars show standard deviation.

**Table 1 ijerph-17-03349-t001:** Sample description—baseline.

	**Total (n = 159)**	**Intervention (n = 69)**		**Enhanced Control (n = 20)**	**Regular Control (n = 70)**	**F Value ANOVA (df)**	***p* Value**
Child age (months) Mean (SD)	37.46 (23.01)	40.68 (22.98)		39.15 (23.18)	33.80 (22.80)	1.742 (2155)	0.179
Parental cigarettes/day Mean (SD)	15.02 (9.59)	14.56 (10.01)		16.75 (8.13)	14.97 (9.62)	0.403 (2156)	0.669
		**Total (n = 159)**	**Intervention (n = 69)**	**Enhanced control (n = 20)**	**Regular control (n = 70)**	**Chi square**	***p* Value**
Child gender	Female	82 (51.6%)	40 (58.0%)	12 (60.0%)	30 (42.9%)	3.829	0.147
Mother’s education	No academic degree	50 (30.8%)	26 (38.0%)	6 (30.0%)	18 (26.1%)	2.205	0.332
	Academic degree	106 (67.9%)	42 (61.8%)	14 (70.0%)	50 (73.5%)		
Father’s education	No academic degree	82 (51.9%)	38 (56.0%)	11 (55.0%)	32 (45.7%)	0.876	0.645
	Academic degree	69 (43.9%)	28 (41.8%)	8 (40.0%)	33 (47.1%)		
Socioeconomic status (self-reported family income)	Above average	68 (43.0%)	32 (47.1%)	7 (35.0%)	29 (41.4%)	7.311	0.293
	Average	49 (31.0%)	20 (29.4%)	5 (25.0%)	24 (34.3%)		
	Below average	36 (22.8%)	12 (17.6%)	7 (35.0%)	17 (24.3%)		

**Table 2 ijerph-17-03349-t002:** Mean Parental Perceptions of Exposure at T1 (study beginning) and T2 (study end) by study groups.

	PPE at T1	PPE at T2	Mean PPE Change	Paired t Test (df)	*p* Value
Intervention (n = 69)	88.13 ± 18.35 (n = 67)	94.76 ± 17.55 (n = 62)	7.02 (17.81) (n = 61)	−3.077 (60)	0.003
Enhanced Control (n = 20)	84.40 ± 21.39 (n = 15)	89.44 ± 18.98 (n = 16)	13.58 (15.42) (n = 12)	−3.052 (11)	0.011
Regular control (n = 70)	-	87.00 ± 19.52 (n = 66)	-		-
Total with repeated PPE measures (n = 73)	86.51 ± 19.34	94.60 ± 17.60	8.10 (17.51)	−3.950 (72)	<0.001

**Table 3 ijerph-17-03349-t003:** Change in behavioural outcomes by study end N (% of families in which the change was reported).

	Intervention	Control	Enhanced Control
Home smoking ban	4 (5.8%)	1 (1.4%)	0
Car smoking ban	1 (1.4%)	0	0
Quit smoking	9 (13.0%)	5 (7.1%)	0
Reduced smoking	2 (2.9%)	5 (7.1%)	1 (5.0%)
Smoke less around children/close balcony door when smoking	11 (15.9%)	1 (1.4%)	1 (5.0%)
Any positive change	39.1%	17.1%	10.0%

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
