# Peer review of "Changing Exposure Perceptions: A Randomized Controlled Trial of an Intervention with Smoking Parents"

_ijerph, 2020, doi:10.3390/ijerph17103349_

Round 1
Reviewer 1 Report
Overall, an interesting study aimed at defining whether the increased perception of their children's exposure to cigarette smoke can affect parental smoking behavior.
Lines 27-28 can be combined into one sentence.
Line 43: The term “partial home ban” must be defined.
Line 56: It is not clear which “tool” was developed. Once again it will be helpful if this is defined.
Line 125: It is not clear whether the hair sample at the beginning and the end of the study was obtained from the “same” child.
Line 128: Change sentence to “JHU second hand smoke exposure lab, which”
Table 1: Define SES.
Figure 1: Why is standard deviation not shown in the figure?
Line 171: Do the % values indicate the percentage of parent who quit smoking?
Author Response
We thank the reviewers for their quick review and helpful comments, and detail our responses below.
REVIEWER 1
Comments and Suggestions for Authors
Overall, an interesting study aimed at defining whether the increased perception of their children's exposure to cigarette smoke can affect parental smoking behavior.
Lines 27-28 can be combined into one sentence. These lines were combined as suggested.
Line 43: The term “partial home ban” must be defined.
We added a definition of partial home bans as follows: “for example restricting smoking to certain areas, or refraining from smoking when children are present”(line 43).
Line 56: It is not clear which “tool” was developed. Once again it will be helpful if this is defined.
We have now specified that a questionnaire was developed, which was subsequently applied in the study. The questionnaire was developed and validated at an earlier stage of the research, published here (1).
Line 125: It is not clear whether the hair sample at the beginning and the end of the study was obtained from the “same” child.
We added the information that both hair samples were taken from the same child to allow comparison (line 127).
Line 128: Change sentence to “JHU second hand smoke exposure lab, which”
This was corrected.
Table 1: Define SES.
We added a definition of SES, which is in fact self-reported family income (average, above average or below average) relative to the national average household income.
Figure 1: Why is standard deviation not shown in the figure?
Standard deviation error bars have now been added to the figure.
Line 171: Do the % values indicate the percentage of parent who quit smoking?
Indeed the % values indicate the percentage of families in which at least one parent reported having quit smoking by the end of the study. We have added a table for clearer presentation of these results (Table 3).

Reviewer 2 Report
1) Brief and succinct Introduction
2) In general, methods were clearly described
3) Statistical analysis (line 131) - mention normality analysis that was undertaken
4) Line 114 - Authors mentioned PPE change scores were computed. But this was not reflected in the results presented.
5) Process evaluation (line 138) - this is not a complete process evaluation analysis, merely show of compliance or adherence to the program
6) Table 1 - provide explanation on what is meant by SES as footnote
7) Table 2 - it is more meaningful to compare the PPE change scores (re: my comment #4) instead of comparing PPE at T2.
8) 3.2 Behavioral outcomes - it is difficult to visualize this without a figure/table.
Author Response
We thank the reviewers for their quick review and helpful comments, and detail our responses below.
REVIEWER 2
1) Brief and succinct Introduction
2) In general, methods were clearly described
3) Statistical analysis (line 131) - mention normality analysis that was undertaken
This information has been added to the Statistical analysis section (line 133). Normality was assessed via q-q plot and histogram.
4) Line 114 - Authors mentioned PPE change scores were computed. But this was not reflected in the results presented.
Mean PPE change over the course of the trial was 8.1 points, (SD 17.5). While the mean change was greater in the enhanced control group, this difference was not significant. This was added to the manuscript (Table 3).
5) Process evaluation (line 138) - this is not a complete process evaluation analysis, merely show of compliance or adherence to the program
The heading has been changed accordingly to Programme adherence.
6) Table 1 - provide explanation on what is meant by SES as footnote
SES has now been defined in Table 1. It refers to self-reported family income (average, above average or below average) relative to the national average household income.
7) Table 2 - it is more meaningful to compare the PPE change scores (re: my comment #4) instead of comparing PPE at T2.
The PPE change score (same as mean difference in Table 2) was not significantly different between the two groups. We now present both comparisons. However we find it important to compare PE at T2 for all groups, as the control group was only assessed at study end and does not have PPE change score.
8) 3.2 Behavioral outcomes - it is difficult to visualize this without a figure/table.
We agree with the reviewer and a table was added presenting changes in behavioural outcomes.
- Myers V, Shiloh S, Rosen LJ. Parental perceptions of children's exposure to tobacco smoke: Development and validation of a new measure. BMC Public Health. 2018;18(1):1031.
